# Brønsted acid-catalysed enantioselective construction of axially chiral arylquinazolinones

Yong-Bin Wang[1,2], Sheng-Cai Zheng[1], Yu-Mei Hu[1] & Bin Tan[1]

The axially chiral arylquinazolinone acts as a privileged structural scaffold, which is present in a large number of natural products and biologically active compounds as well as in chiral ligands. However, a direct catalytic enantioselective approach to access optically pure arylquinazolinones has been underexplored. Here we show a general and efficient approach to access enantiomerically pure arylquinazolinones in one-pot fashion catalysed by chiral phosphoric acids. A variety of axially chiral arylquinazolinones were obtained in high yields with good to excellent enantioselectivities under mild condition. Furthermore, we disclosed a method for atroposelective synthesis of alkyl-substituted arylquinazolinones involving Brønsted acid-catalysed carbon–carbon bond cleavage strategy. Finally, the asymmetric total synthesis of eupolyphagin bearing a cyclic arylquinazolinone skeleton was accomplished with an overall yield of 32% in six steps by utilizing the aforementioned methodology.

[1] Department of Chemistry, South University of Science and Technology of China, Shenzhen 518055, China. [2] Department of Chemistry, Fudan University, 220 Handan Road, Shanghai 200433, China. Correspondence and requests for materials should be addressed to B.T. (email: tanb@sustc.edu.cn).

The axially chiral arylquinazolinones constitutes a privileged structural scaffold found in a large number of natural products and biologically active compounds as exemplified by those shown in Fig. 1a (refs 1-8). This motif represents a well-known class of therapeutics that displays hypnotic, anxiolytic, anticonvulsant and antitumor effects. For example, eupolyphagin[1] and asperlicins[2] are known as potent cholescystokinin antagonists; Erastin[3] is an antitumor agent used for targeting selective tumor cells containing oncogenic RAS, which yields ferroptosis by changing the mitochondrial voltage-dependent anion channel gating thus allowing cations into the mitochondria, resulting to the release of oxidative species. In addition, this motif could also be used as chiral ligand for asymmetric catalysis[9,10].

The significance of these privileged axially chiral skeletons has led to a great demand for efficient synthetic methods, particularly those producing enantiomerically pure arylquinazolinones. In this regard, there have only existed a few transformations for the construction of these structural skeletons, which usually employ either enantiopure starting materials or conventional resolution to construct the optically pure arylquinazolinone derivatives[11,12]. Levacher and co-workers developed a divergent stereoselective synthesis of bridged arylquinazolinones by using the popular Meyers' diastereo-selective lactamization under dehydrating conditions[12]. Miller reported a pioneering approach for accessing the enantioenriched arylquinazolinone via peptide-catalysed atroposelective bromination of preformed quinazolinones, representing the only asymmetric catalytic example (Fig. 1b)[13]. However, the hydroxyl group is necessary to direct the atroposelective bromination, and only alkyl

group is tolerant in the reaction, which restricted the further application for diversity-oriented-synthesis. Although these were elegant and creative strategies towards the synthesis of these skeletons, the direct catalytic enantioselective approach to access optically pure arylquinazolinones was yet to be described and would be of great value with respect to synthetic efficiency and broader substrate scope.

Since the pioneering reports of Akiyama[14] and Terada[15], chiral phosphoric acids have been widely used as organocatalysts in the simultaneous activation of nucleophile and/or electrophile for asymmetric transformations[16–19]. Recently, several elegant examples for asymmetric synthesis of axially chiral compounds[20–26] were reported by using phosphoric acids[27–34]. In this context, Akiyama and co-workers achieved a significant breakthrough by using phosphoric acid to enable the asymmetric atroposelective bromination, providing a useful approach for accessing axially biaryldiols[27]. Recently, they also developed an enantiodivergent synthesis of axially chiral biaryls via asymmetric reductive amination catalysed by a chiral phosphoric acid[28]. Motivated by these successful examples and the previous efforts on phosphoric acid-catalysed asymmetric synthesis of dihydroquinazolinones[35,36] via amidation[37], we envisioned that chiral phosphoric acids may also be capable of facilitating the construction of axially chiral arylquinazolinones enantioselectively by accelerating the imine formation and the intramolecular nucleophilic addition to form hemiaminal followed by the oxidative dehydrogenation (Fig. 1c).

However, several challenges are associated with the direct asymmetric construction of the axially chiral arylquinazolinones

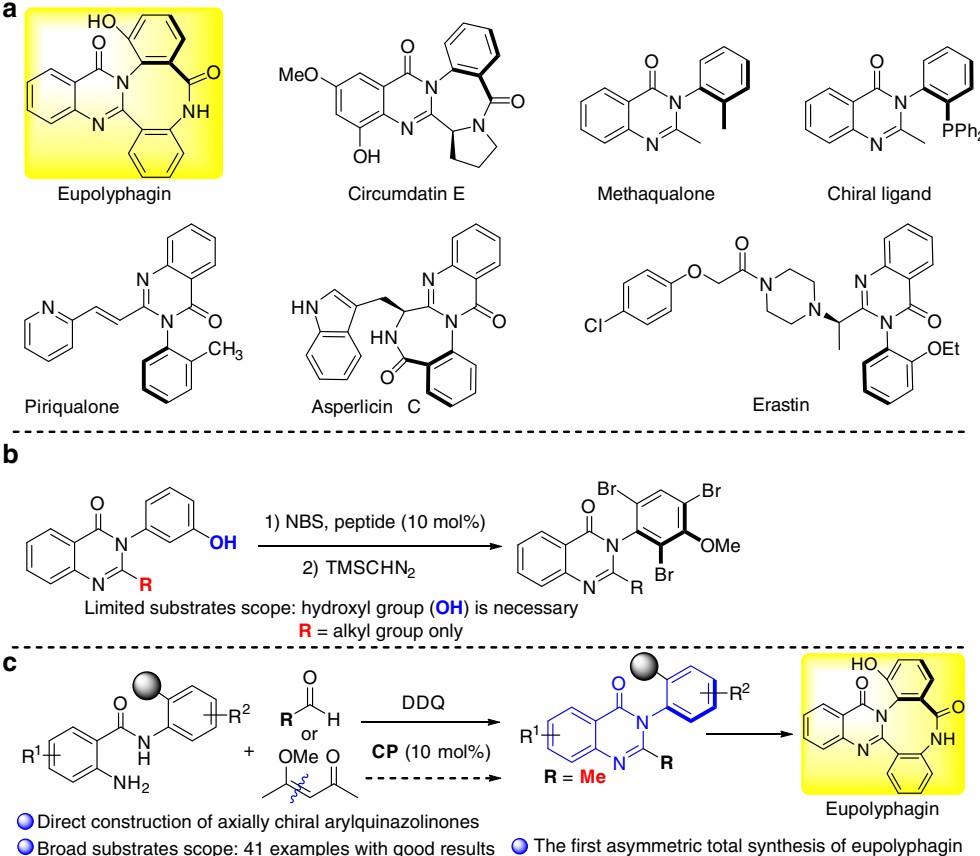

**Figure 1 | Representative arylquinazolineones and catalytic asymmetric construction of arylquinazolinone. (a)** Natural products, biologically active compounds and chiral ligand bearing axially chiral arylquinazolinones. **(b)** Miller's pioneering work. **(c)** Our strategy for atroposelective construction of arylquinazolineones and its application for total synthesis of eupolyphagin.

in a one-pot fashion: The atroposelective construction of axially chiral arylquinazolinones has rarely been investigated in asymmetric catalysis; the previous reports on asymmetric synthesis of dihydroquinazolinones did not investigate the N-aryl anthranilamides; the concomitant control of the stereoselectivities of dihydroquinazolinones bearing a stereogenic carbon center and an axial chirality might be a great challenge; the development of a suitable oxidation condition for preserving the enantiopurity needs to be addressed. As part of our continuous efforts in the asymmetric construction of axially chiral compounds[38] and inspired by the elegant work of Rodriguez and Bonne involving oxidative central-to-axial chirality conversion strategy[39,40], here we describe the results of the investigation on addressing the aforementioned challenges, leading to the phosphoric acid-catalysed highly enantioselective synthesis of axially chiral arylquinazolinones with high enantiopurity and structural diversity (Fig. 1c). In addition, the phosphoric acid-catalysed carbon–carbon bond cleavage for direct atroposelective construction of alkyl-substituted arylquinazolinones under mild condition suggests an approach of great importance to medicinal chemistry and diversity-oriented synthesis.

## Results

**Optimization of reaction conditions.** To validate the feasibility of the hypothesis, our initial investigations were carried out by

using the reaction of N-aryl anthranilamides (**1a**) and benzaldehyde (**2a**) as the model substrates with 10 mol% of phosphoric acid **CP1** in CHCl$_3$ at 0 °C in the presence of 2,3-dichloro-5,6-dicyano-1,4-benzoquinone (DDQ). Despite its high steric hindrance, the reaction proceeded cleanly to give the axially chiral arylquinazolinone (**3a**) with 68% ee, albeit with a low conversion (Table 1, entry 1). This preliminary result obviously demonstrated that the control of the axial chirality of arylquinazolinone by using chiral phosphoric acid-catalysed asymmetric cyclo-condensation and oxidative dehydrogenation is feasible. To improve the reactivity and stereoselectivity, we next turned our attention to evaluate the different phosphoric acid catalysts available. As shown in Table 1, the electron property and steric bulk on the aromatic ring, as well as the backbone displayed remarkable effects on the outcome of the reaction (Table 1, entries 2–8). Fortunately, we found the catalyst **CP3** gave the best yield of 88% and enantioselectivity of 96% ee (Table 1, entry 3). It is worthy of note that the more acidic N-triflylphosphoramides **CP7**, which were first reported by Yamamoto and Nakashima[41], did not improve the results (entry 7). The solvent had an important role in the transformation, and CHCl$_3$ was proved to be the best choice (Table 1, entries 9–12). Expectedly, the presence of 4 Å molecular sieves had a remarkable effect on the enantioselectivity because of the generation of water during the reaction (Table 1, entry 13). Further optimization of the conditions revealed that the reaction was complete to furnish

---

**Table 1 | Optimization of the reaction conditions\*.**

| Entry | Catalyst | Solvent | Yield (%)† | ee (%)‡ |
|---|---|---|---|---|
| 1 | **CP1** | CHCl$_3$ | 21 | 69 |
| 2 | **CP2** | CHCl$_3$ | 58 | 49 |
| 3 | **CP3** | CHCl$_3$ | 88 | 96 |
| 4 | **CP4** | CHCl$_3$ | 22 | 13 |
| 5 | **CP5** | CHCl$_3$ | 22 | 38 |
| 6 | **CP6** | CHCl$_3$ | 58 | 63 |
| 7 | **CP7** | CHCl$_3$ | 11 | 39 |
| 8 | **CP8** | CHCl$_3$ | 48 | -67 |
| 9 | **CP3** | CH$_2$Cl$_2$ | 51 | 67 |
| 10 | **CP3** | CCl$_4$ | 67 | 91 |
| 11 | **CP3** | Toluene | 13 | 81 |
| 12 | **CP3** | EA | 8 | 58 |
| 13§ | **CP3** | CHCl$_3$ | 86 | 74 |
| 14‖ | **CP3** | CHCl$_3$ | 99 | 93 |
| **15¶** | **CP3** | **CHCl$_3$** | **96** | **96** |
| 16# | **CP3** | CHCl$_3$ | 95 | 96 |

\*Unless otherwise specified, the reaction of **1a** (0.1 mmol), **2a** (0.2 mmol), DDQ (31.9 mg, 0.14 mmol), catalyst **CP** (10 mol%) and 4 Å MS was carried out in 4.0 ml solvent at 0 °C for 60 h under Ar.
†Isolated yield.
‡Determined by high-performance liquid chromatography analysis.
§Without 4 Å MS.
‖At 25 °C.
¶The reaction was performed for 96 h.
#DDQ was added after 48 h and then the reaction was performed for additional 96 h.

---

**Table 2 | Substrates scope of *N*-aryl anthranilamides\*.**

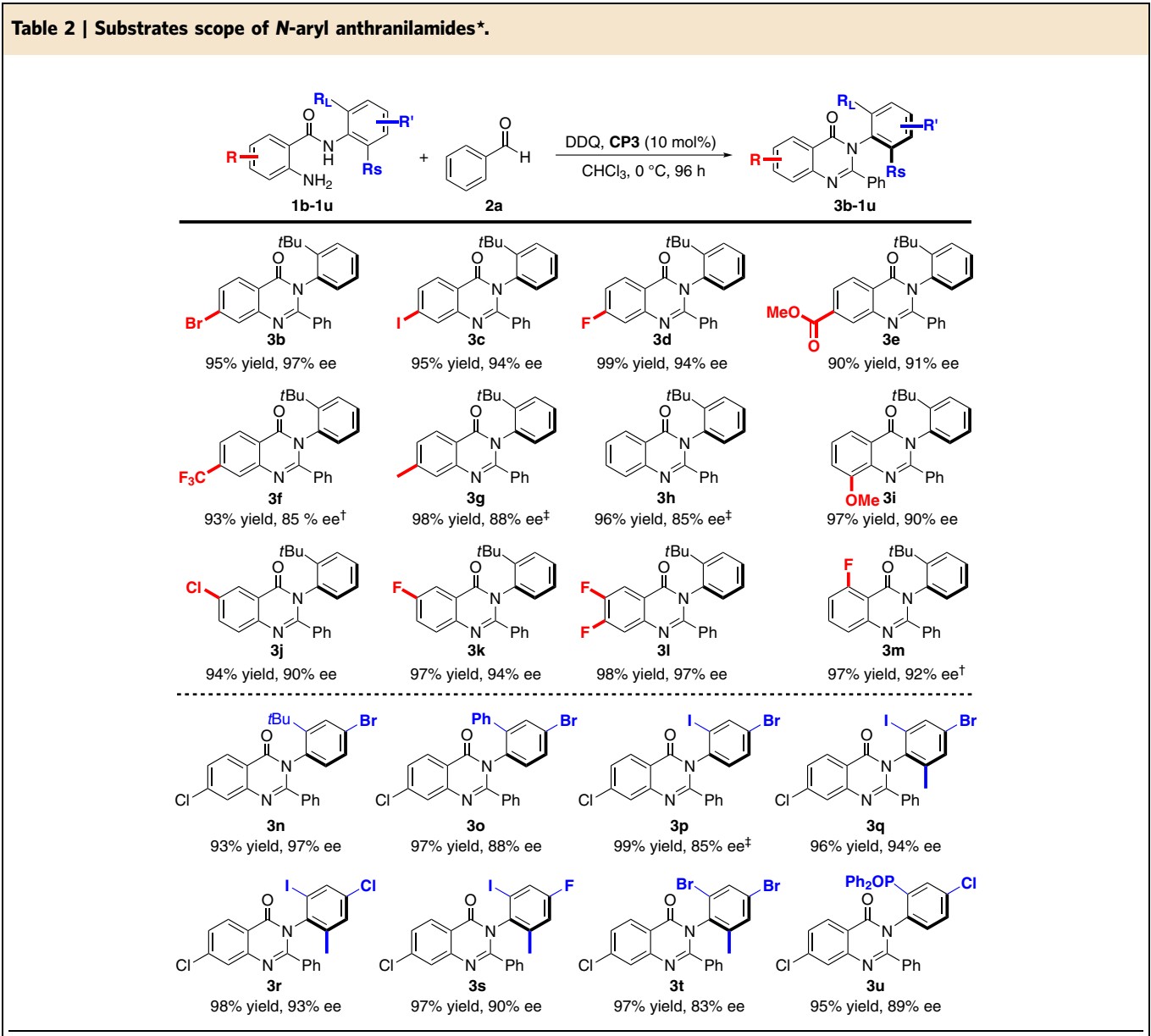

\*Unless otherwise specified, the reaction of **1a** (0.1 mmol), **2a** (0.2 mmol), **DDQ** (31.9 mg, 0.14 mmol), **CP3** (7.0 mg, 10 mol%) and 4 Å MS was carried out in 4.0 ml CHCl₃ at 0 °C for 96 h under Ar.
†At 25 °C.
‡Performed in *c*-Hexane:CHCl₃ = 2:1.

**3a** in 96% isolated yield with 96% ee (Table 1, entry 15). It was demonstrated that the oxidant DDQ could be added after the first cyclization reaction was accomplished without any effect on the reactivity and stereoselectivity (Table 1, entry 16). The absolute configuration of **3a** was determined as being (*aR*) by X-ray diffraction analysis (See Supplementary Fig. 1, CCDC 1509246) and those of other products were assigned by analogy.

**Substrate scope**. With the optimal reaction conditions in hand, we set out to explore the substrate generality of this transformation. First, we evaluated various substituted anthranilamides (**1b–1m**). Most reactions reached completion within 96 h and gave axially chiral arylquinazolinones (Table 2, **3b–3m**) in high yields (90–99%) with good to exellent enantioselectivities (85–97% ee). The results revealed that both the position and electronic property of the substituents on the aromatic ring have a slight effect on the reaction efficiency and enantioselectivity of this transformation. With respect to the *N*-aryl group, it is noteworthy to point out that the ortho group is not only restricted to *tert*-butyl (*t*Bu) group, and the bromo, iodo, phenyl or diphenyl phosphine oxide (Ph₂OP) group at this position could also be suitable substrates that yielded the desired products (Table 2, **3n–3u**) with good to excellent enantiocontrol (83–95% ee). Notably, the presence of Cl, Br or I substituent in the aryl-quinazolinone derivatives obtained is very important for setting up a compound library due to the high reactivity in many transition-metal-catalysed reactions[42].

Encouraged by these results, we, therefore, expanded the generality of the reaction with regard to the variation of the aldehydes. Various aromatic aldehydes were applicable and afforded the desired products with good yields and excellent enantioselectivities (Table 3). It was shown that the position and the electronic nature of the substituents on the aromatic ring have limited effect on the stereoselectivity. For example, aldehydes bearing electron withdrawing (R = F, Cl, Br) or electron-donating groups (R = Me, OMe) at different positions (para, meta, ortho) of the phenyl ring reacted efficiently to afford the corresponding

**Table 3 | Substrates scope of aldehydes\*.**

*Unless otherwise specified, the reaction of **1a** (0.1 mmol), **2a** (2.0 mmol), DDQ (1.4 equiv), **CP3** (10 mol%) and 4 Å MS was carried out in 4.0 ml CHCl₃ at 0 °C for 96 h under Ar.
†At 25 °C.

products **4a–4j** with 63–98% yields and 85–95% ee. Moreover, the 2-furaldehyde and 2-naphthaldehyde could be well tolerated without obviously affecting the reaction results (**4k** and **4l**). When aliphatic aldehydes were investigated, the reaction proceeded very messy under the optimized conditions. To our delight, the expected arylquinazolinone (**4m**) was produced when DDQ was added after the first step was completed. In the meanwhile, an unexpected product **4n** was obtained with excellent enantioselectivity (92% ee) using phenyliodinediacetate as oxidant. Disappointedly, although we tried our best to optimize the reaction conditions for linear aliphatic aldehydes, no good results were obtained.

**C–C bond cleavage strategy**. To further expand the generality to get alkyl-substituted arylquinazolinone, we were interested in a promising strategy via phosphorous acid-catalysed cyclocondensation and selective C–C bond cleavage to meet this challenge. As we all known, C–C bond cleavage is a topic of significant importance in organic synthesis[43,44]. Despite significant advances in the past decades, it is still a big challenge to selectively cleave unstrained C–C bond by using an organocatalyst. In this regard, Zhou and co-workers[45] developed an elegant approach to deliver quinazolinones involving organocatalytic reactions of ketoesters/diketones with 2-aminobenzamides and selective C–C bond cleavage. Inspired by

this beautiful work, we envisioned that the axially chiral arylquinazolinone might be produced by using the reaction of N-aryl anthranilamides (**1h**) with ketoester or diketone under the catalysis of chiral phosphoric acids. To our disappointment, almost no desired product was obtained when ketoester was tested. After substantial optimization (for optimized details, see Supplementary Table 1), the axially chiral arylquinazolinone **6a** was obtained with good results by treating the reaction with 4-methoxypentenone **5a** and **1h** catalysed by a more acidic N-triflylphosphoramides **CP9** at 60 °C in a mixture of c-hexane and CHCl₃ in the presence of MgSO₄ (Table 4, product **6a**). The absolute configuration of **3a** was determined to be (aR) by X-ray diffraction analysis (Supplementary Fig. 2, CCDC 1509247). Having identified the optimized conditions, we proceeded to investigate the substrate scope of the reaction. Several substituted N-aryl anthranilamides were successfully applied in the reaction with 4-methoxypentenone (Table 4). The corresponding axially chiral methyl-substituted arylquinazolinones **6a–6f** were isolated with good chemical yields (75–95%) and enantioselectivities (83–95% ee). It should be noted that other diketone derivatives produce the corresponding products with poor results, which shows the limitation of the current method. However, the resultant product **6f** can be further transformed into other diversely functionalized axially chiral arylquinazolinones **7a–7c**

**Table 4 | Further expansion of the generality involving C–C bond cleavage strategy\*.**

\*Unless otherwise specified, the reaction of **1h** (0.1 mmol), **5a** (0.2 mmol), **CP9** (8.0 mg, 10 mol%) and MgSO₄ (6.0 mg) was carried out in 4.0 ml mixed solvent (*c*-hexane:CHCl₃ = 1:1) at 60 °C for 96 h under Ar.

**Figure 2 | Asymmetric total synthesis of eupolyphagin.** The natural product eupolyphagin was effectively synthesized in six steps from easily available starting material with overall yield of 32% in 95% ee.

without any erosion of enantioselectivities, thus expanding the substrate scope and compensating the limitation to some extent.

**Asymmetric total synthesis of eupplyphagin**. The efficient construction of a series of functionalized axially chiral arylquinazolinones motivated our investigation of the asymmetric total synthesis of eupolyphagin (Fig. 2). Our primary concept is to construct the key axially chiral intermediate **8c** by using the above-mentioned approach and followed by construction of

eight-member lactam **8e** via palladium-catalysed carbonylation of aryl halides. We started our investigation from the optimization of the construction of the key axially chiral intermediate **8c**. After great efforts, the corresponding product was obtained with moderate enantioselectivity by treating the reaction with **8b** and 2-FmocNHPhCHO catalysed by (*S*)-**CP3** at 40 °C (other *N*-protected aldehydes and anthraniamides gave poor results. For details, see Supplementary Table 2). Fortunately, the enantiopurity of **8c** can be enriched to 95% ee by using recrystallization from dichloromethane (DCM)/ethyl acetate (EA). The free amine

**Figure 3 | Control experiments proposed reaction processes.** (**a**) The intermediate **9a** was obtained as a single diastereoisomer (dr > 20:1) with good enantioselectivity. (**b**) Two diastereoisomers (dr = 20:1) of **9b** was isolated. (**c**) The two diastereoisomers of **9c** were inseparable and the following oxidative transformations gave **3p** with slightly decrease of enantioselectivity. (**d**) The isolated intermediates **9d** could be transformed to **6a** successfully. (**e**) Proposed reaction processes.

**8d** was obtained by removing the Fmoc protection group in excellent yield without any effect on the enantioselectivity. Although palladium-catalysed carbonylation of aryl halides provided a simple method that generated a range of carboxylic acid derivatives, there was only one report for the construction of eight-member lactam[46]. Considering the complicated catalyst and high carbon monoxide pressure that may have a strong effect on the conversion of aryl iodide **8d** to the key intermediate **8e**, we selected Arndtsen's modified procedure[47]. Our preliminary investigation by using $tBu_3P$ as ligand and TBACl as chloride source afforded the desired **9e** with moderate yield. Further optimization (for optimized details, see Supplementary Table 3) resulted in an improved protocol which could provide **8e** in satisfactory isolated yield (95%) without any effect on enantioselectivity. Finally, the methyl protecting group was removed by using $BBr_3$ to obtain eupolyphagin as a white solid in 98% yield with 95% ee[48]. It should be emphasized that it represents the asymmetric total

synthesis of this axially chiral natural product and the overall yield for the six steps is 32%.

**Controlled experiments and proposed reaction processes.** In order to get some insights of the reaction mechanism, we have conducted several control experiments (Fig. 3). In the beginning, we performed a control experiment in the absence of oxidant and successfully isolated the key aminal intermediate **9a** in 90% yield and 96% ee and only one diastereoisomer was detected (dr > 20:1), indicating that the double chirality was controlled completely. The following oxidation of **9a** with DDQ afforded the desired product **3a** with excellent yield and 94% ee (Fig. 3a). However, when we isolated another intermediate **9b**, the crude-NMR clearly showed that the dr of **9b** was 20:1. The two diastereoisomers yielded the corresponding axially chiral product **3t** with opposite configuration (Fig. 3b). In addition, when we attempted to isolate the intermediate **9c**, we found that the two

diastereoisomers (dr = 3.6:1 in CDCl$_3$) cannot be separated on silica gel chromatography and only two peaks were detected by chiral high-performance liquid chromatography analysis. The transformation of this intermediate afforded the final product **3p** with excellent yield and 81% ee (Fig. 3c). Based on these results, we deduced that the C–N bond can rotate freely in the intermediate **9c** and the central chirality can be efficiently transferred into axial chirality during the oxidation. These results also indicated that the axial chirality of the final product was controlled in the cyclization step and the control of the central chirality was the key point for achieving good enantioselectivity. As for the synthesis of alkyl substituted arylquinazolinone, we have isolated the enamine **9d** and failed to separate the cyclization intermediate due to the unstability. The final product **6a** was isolated with 50% yield and 77% ee under standard condition (Fig. 3d). On the basis of above-mentioned control experiments and previous reports[35–37], we proposed tentative reaction processes. As shown in Fig. 3e, these reactions probably go through the typical Brønsted acid-catalysed aminal formation process. Initially, imines or enamines were generated from the condensation of **1** with aldehyde or 4-methoxypentenone, respectively. Then *N*,*N*-aminal cyclization intermediates were produced via intramolecular amidation. Finally, the desired arylquinazolinones were obtained by dehydrogenation in the presence of an oxidant or the C − C bond cleavage under acidic condition[48]. It is clear that the enantioselectivity is determined during the step of intramolecular nucleophilic attack under the catalysis of chiral Brønsted acid. At the current stage, the asymmetric induction and the chirality transfer from central to axial chirality remains unclear, which is ongoing in our laboratory.

## Discussion

We have successfully developed a general and efficient approach to access enantiomerically pure arylquinazolinones catalysed by chiral phosphoric acids. A series of axially chiral arylquinazolinones with potential biological activities were obtained in high yields (up to 99%) with good to excellent enantioselectivities (up to 97% ee) under mild conditions. To expand the reaction generality, we further disclosed a method of great importance for the atroposelective construction of alkyl-substituted arylquinazolinones using Brønsted acid-catalysed carbon–carbon bond cleavage strategy. Applying this methodology, the asymmetric total synthesis of axially chiral natural product eupolyphagin was achieved with an overall yield of 32% in six steps. We anticipate that this promising strategy could be applied to the synthesis of other natural products and the axially chiral arylquinazolinones may have further potential applications in asymmetric catalysis and drug discovery.

## Methods

**Procedure for the enantioselective synthesis of 3 and 4a–4l.** To a dry Schlenk tube (10 ml), 400 mg activated 4 Å MS (molecular sieves) was added and then the MS was reactivated under reduced pressure for 15 min. After the tube was cooled down, *N*-aryl anthranilamides **1** (0.10 mmol), **CP3** (7.0 mg, 0.01 mmol), DDQ (31.8 mg, 0.14 mmol) and anhydrous CHCl$_3$ (4.0 ml) was added under Ar. The resulting mixture was stirred for 10 min at 0 °C, and then aromatic aldehyde **2** (0.2 mmol) was added in one portion. After stirred for 96 h at 0 °C, the mixture was purified by flash column chromatography on silica gel (gradient elution with PE/EA) to give the pure product.

**Procedure for the enantioselective synthesis of 6.** To a 10 ml dry Schlenk tube, *N*-aryl anthranilamides **1h**, **1m** or **1v-1y** (0.10 mmol), **5a** (22.8 mg, 0.2 mmol), MgSO$_4$ (6.0 mg), **CP9** (8.0 mg, 0.01 mmol) and anhydrous *c*-hexane/CHCl$_3$ = 1/1 (4.0 ml) was added under Ar. The resulting solution was heated to 60 °C. After stirred for 96 h, the solution was cooled down to room temperature and then purified by flash column chromatography on silica gel (gradient elution with PE/EA) to give the pure product.

For NMR spectra of the new compounds in this article, see Supplementary Figs 3–209. Full experimental details and characterization of compounds are given in Supplementary Notes 1–5.

**Data availability.** The X-ray crystallographic coordinates for structures reported in this Article have been deposited at the Cambridge Crystallographic Data Centre (CCDC), under deposition numbers CCDC 1509246, CCDC 1509247. Crystal data and structure refinement for **3a** and **6f** were displayed in Supplementary Tables 4 and 5. These data can be obtained free of charge from The Cambridge Crystallographic Data Centre via http://www.ccdc.cam.ac.uk/data_request/cif.

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

## Acknowledgements

We are thankful for the financial support from the National Natural Science Foundation of China (Nos. 21572095), Shenzhen special funds for the development of biomedicine, internet, new energy, and new material industries (JCYJ20150430160022510). B.T. thanks the Thousand Young Talents Program for financial support. Dedicated to prof Scott J. Miller for his great contribution on construction of axially chiral compounds via peptide catalysis.

## Author contributions

Y.-B.W. performed experiments and prepared the Supplementary Information. S.-C.Z. and Y.-M.H. helped with characterizing some new compounds. B.T. conceived and directed the project and wrote the paper.

## Additional information

**Competing interests:** The authors declare no competing financial interests.

**Publisher's note**: 

