## [Peer review file · Nature Communications]

REVIEWERS' COMMENTS:

Reviewer #1 (Remarks to the Author):

Tan and co-workers describe in this manuscript enantioselective synthesis of axially chiral arylquinazoines by means of chiral phosphoric acid.

Recently, enantioselective synthesis of axially chiral compounds has attracted much attention of synthetic organic chemists.

Based on the control experiments, the authors confirmed that the enantioselectivity was determined by the step of intramolecular amidation of imines.

This is a nice piece of work and this reviewer recommends publication of this manuscript in Nature Communications in the present form.

Reviewer #2 (Remarks to the Author):

The axially chiral arylquinazolinones are valuable structural units in pharmaceutical agents, and could be used as chiral ligand for asymmetric catalysis. Chiral phosphoric acids have been extensively used in enantioselective catalysis. The manuscript describes the chiral phosphoric acids-catalyzed methodology for novel axially chiral arylquinazolinones with high enantiopurity and structural diversity. Furthermore, a new method involving phosphoric acid-catalysed carbon-carbon bond cleavage for atroposelective construction of alkyl substituted arylquinazolinones was also disclosed. A large-scale experiment documents the applicability of this reaction and a reasonable transition state model is put forth as well by control experiments. The nice work seems to have been well conducted with sufficient details, and can be of utility to researchers interested in the organic chemistry and medicinal chemistry. Thus I recommend the publication of this manuscript in this journal after following modification is made:

Many products were obtained only < 90% ee using this method. Can enantiopurity of all these products be enriched to >95% ee by using recrystallization?

Reviewer #3 (Remarks to the Author):

The authors report a remarkable approach to enantiopure, axially chiral arylquinazolinones that involves reacting an aldehyde with an amido aniline in the presence of an oxidant. The reaction is catalyzed by a chiral phosphoric acid and gives the corresponding products in high enantioselectivity. Mechanistic studies reveal that the reaction proceeds via diastereoselective and enantioselective aminal formation followed by oxidation to give the axially chiral product. A very elegant application of the method in the synthesis of Eupolyphagin is also described. The language needs some improvement.

Our responses to the Reviewers

Reviewer #1 (Remarks to the Author):

Tan and co-workers describe in this manuscript enantioselective synthesis of axially chiral arylquinazoines by means of chiral phosphoric acid. Recently, enantioselective synthesis of axially chiral compounds has attracted much attention of synthetic organic chemists. Based on the control experiments, the authors confirmed that the enantioselectivity was determined by the step of intramolecular amidation of imines. This is a nice piece of work and this reviewer recommends publication of this manuscript in Nature Communications in the present form.

Our response: We deeply appreciate these encouraging comments of the referee and sincerely thank the referee for strong support for publication in *Nature Communications*.

Reviewer #2 (Remarks to the Author):

The axially chiral arylquinazolinones are valuable structural units in pharmaceutical agents, and could be used as chiral ligand for asymmetric catalysis. Chiral phosphoric acids have been extensively used in enantioselective catalysis. The manuscript describes the chiral phosphoric acids-catalyzed methodology for novel axially chiral arylquinazolinones with high enantiopurity and structural diversity. Furthermore, a new method involving phosphoric acid-catalysed carbon-carbon bond cleavage for atroposelective construction of alkyl substituted arylquinazolinones was also disclosed. A large-scale experiment documents the applicability of this reaction and a reasonable transition state model is put forth as well by control experiments. The nice work seems to have been well conducted with sufficient details, and can be of utility to researchers interested in the organic chemistry and medicinal chemistry. Thus I recommend the publication of this manuscript in this journal after

following modification is made:

Many products were obtained only < 90% ee using this method. Can enantiopurity of all these products be enriched to >95% ee by using recrystallization?

Our response: We very much appreciate these supportive comments of the referee and sincerely thank the referee for bring the issue to our attention. The products as solid showed in the following table were obtained < 90% ee. We conducted several experiments to enrich their ees by using recrystallization except **3f**, **4j**, **4k** and **6e** which were obtained as oil. The solid was dissolved in refluxing solvent and then the solution was cooled down to room temperature slowly. Until no more solid precipitated, the mixture was filtered and the ees of the filtrate and filter cake (solid) were determined by using chiral HPLC. The filter cake afforded **3u**, **6b** and **6c** with moderate to good yield and excellent ee (>95%). However, in the case of **3h**, **6a** and **6d**, the products with satisfactory ee (>95) and good yield were existing in the filtrate. Only oil formed when the solution of **4i** in PE or *n*-Hexane was cooled down and the ees of the precipitated oil has

no difference with that of the filtrate. Similarly, our attempt to enrich the ee of **3p** through recrystallization in several solvent systems was failed. The related results have been added to the revised Supplementary Information.

Compounds	Property	Recrystallization			
		Solvent	Obtained Phase	Yield (%)	Ee (%)
3h	solid	PE	Filtrate	65	97
3u	solid	PE/DCM	Filter Cake	51	>99
6a	solid	PE	Filtrate	84	97
6b	solid	PE	Filter Crystal	64	99
6c	solid	PE	Filter Crystal	86	98
6d	solid	PE	Filtrate	87	98
4i	solid	PE or n -Hexane	Failed	--	--
3p	solid	PE/EA, PE/DCM or n -Hexane/Et ₂ O	Failed	--	--

Reviewer #3 (Remarks to the Author):

The authors report a remarkable approach to enantiopure, axially chiral arylquinazolinones that involves reacting an aldehyde with an amido aniline in the presence of an oxidant. The reaction is catalyzed by a chiral phosphoric acid and gives the corresponding products in high enantioselectivity. Mechanistic studies reveal that the reaction proceeds via diastereoselective and enantioselective amination followed by oxidation to give the axially chiral product. A very elegant application of the method in the synthesis of Eupolyphagin is also described. The language needs some improvement.

Our response: We very much appreciate these comments of the referee and sincerely thank the referee for recommending for publication in *Nature Communications*. According to the referee's valuable suggestion, we have carefully modified the manuscripts and asked one of my colleagues to polish English.